# Comparative Transcriptome Analysis Reveals an Efficient Mechanism for α-Linolenic Acid Synthesis in Tree Peony Seeds

**DOI:** 10.3390/ijms20010065

**Published:** 2018-12-24

**Authors:** Qingyu Zhang, Rui Yu, Daoyang Sun, Md Mahbubur Rahman, Lihang Xie, Jiayuan Hu, Lixia He, Aruna Kilaru, Lixin Niu, Yanlong Zhang

**Affiliations:** 1College of Landscape Architecture and Arts, Northwest A&F University, Yangling 712100, China; zhangqingyu@nwafu.edu.cn (Q.Z.); yurui@nwafu.edu.cn (R.Y.); sundaoyang@nwafu.edu.cn (D.S.); xielihang@nwafu.edu.cn (L.X.); hujiayuan@nwafu.edu.cn (J.H.); 2Department of Biological Sciences, East Tennessee State University, Johnson City, TN 37614, USA; rahmanm@etsu.edu; 3Gansu Forestry Science and Technology Extend Station, Lanzhou 730046, China; helixia_lz@sina.com

**Keywords:** transcriptome, α-linolenic acid, fatty acid desaturase, tree peony, essential fatty acids, triacylglycerol

## Abstract

Tree peony (*Paeonia* section *Moutan* DC.) species are woody oil crops with high unsaturated fatty acid content, including α-linolenic acid (ALA/18:3; >40% of the total fatty acid). Comparative transcriptome analyses were carried out to uncover the underlying mechanisms responsible for high and low ALA content in the developing seeds of *P. rockii* and *P. lutea*, respectively. Expression analysis of acyl lipid metabolism genes revealed upregulation of select genes involved in plastidial fatty acid synthesis, acyl editing, desaturation, and triacylglycerol assembly in seeds of *P. rockii* relative to *P. lutea*. Also, in association with ALA content in seeds, transcript levels for fatty acid desaturases (SAD, FAD2, and FAD3), which encode enzymes necessary for polyunsaturated fatty acid synthesis, were higher in *P. rockii* compared to *P. lutea*. Furthermore, the overexpression of *PrFAD2* and *PrFAD3* in *Arabidopsis* increased linoleic and ALA content, respectively, and modulated the final ratio 18:2/18:3 in the seed oil. In conclusion, we identified the key steps and validated the necessary desaturases that contribute to efficient ALA synthesis in a woody oil crop. Together, these results will aid to increase essential fatty acid content in seeds of tree peonies and other crops of agronomic interest.

## 1. Introduction

Tree peony belongs to the section *Moutan* DC, genus *Paeonia*, and family Paeoniaceae and is endemic to China. Recently, tree peony has attracted wide research interests due to the high amount of polyunsaturated fatty acids (PUFAs, >90%) in its seeds [1]. The α-linolenic acid (ALA, 18:3) content of peony seed oil (>40%) is higher than that of conventional vegetable oils derived from corn (1%), soybean (8%), and rapeseed (11%) [2]. ALA is an essential omega-3 fatty acid (FA) for human health and nutrition, as it cannot be synthesized by humans and can only be obtained from diet [3]. Thus, ALA plays an important role in prevention of heart diseases, hypertension, diabetes, and obesity [4,5]. Identifying and engineering plants to generate high levels of omega-3 FAs is a pertinent approach to meet the increasing dietary needs of humans [6]. To this extent, we aimed to elucidate the mechanisms by which tree peony species accumulate ALA in its seeds and develop tree peony species as an alternative source for the production of omega-3 FAs.

In plants, FA synthesis in plastid and triacylglycerol (TAG) assembly in the endoplasmic reticulum (ER) are the two major events in storage lipid accumulation. Typically, FA biosynthesis is catalyzed by the FA synthase complex and requires several steps to synthesize palmitoyl-acyl carrier protein (16:0-ACP) from acetyl-CoA. Palmitoyl-ACP can be further elongated to stearoyl-ACP (18:0-ACP) by the addition of two carbon atoms from acetyl-CoA [7]. In plastid, most of the 18:0-ACP is desaturated to oleoyl-ACP (18:1-ACP) by a soluble stearoyl-ACP desaturase (SAD) [8]. Acyl-ACP thioesterases, mainly FATA, and FATB hydrolyze acyl-ACPs, mostly oleic acid (OA, 18:1), and a small amount of palmitic acid (PA, 16:0) and stearic acid (SA, 18:0) to release free FAs in the plastid stroma [9,10]. These FAs are activated to acyl-CoAs before entering into ER where they can be incorporated into membrane glycerolipids and storage lipids by sequential esterification of glycerol-3-phosphate (G3P) [9]. First, acylation of G3P is catalyzed by the glycerol-3-phosphate acyltransferase (GPAT) at *sn-1* to form lysophosphatidic acid (LPA), followed by a second acylation by LPA acyltransferase (LPAAT) at *sn-2* to produce phosphatidic acid (PA). Third, acylation requires dephosphorylation of PA by a PA phosphatase (PAP) to generate diacylglycerol (DAG), which can be used as a substrate for Kennedy pathway by diacylglycerol acyltransferase (DGAT) to produce TAG [11]. TAG can also be formed by the transfer of acyl groups from phosphatidylcholine (PC) to DAG by the phospholipid: diacylglycerol acyltransferase (PDAT) [12,13].

In mature seeds, desaturation of OA to linoleic acid (LA,18:2) and subsequently to ALA are catalyzed by two ER-localized FA desaturases (FAD2 and FAD3), respectively [14,15]. These desaturases use phospholipids as acyl substrates, and NADH, NADH-cytochrome *b_5_* reductase and Cyt *b_5_* as electron donors [8]. As membrane-bound proteins, they contain three conserved histidine (His) clusters that contain His boxes comprising eight His residues [16,17]. Desaturase genes have been characterized in many plant species, including *Arabidopsis* [15,18], soybean [19], rapeseed [20], and cotton [21]. Only a single *FAD2* gene was found in *Arabidopsis* [14], while in most other plant species, FAD2 is encoded by small or large gene families, and their expression can be tissue-specific or constitutive. For instance, the expression of two *FAD2* genes in olive and grape are constitutive in different organs but specific in seeds [22,23]. Similarly, among the three *FAD2* genes in cotton and sunflower, only one of them is seed-specific, while the other two are ubiquitous [24,25]. Omega-3 FAD3 in microsomes, encoded by the *FAD3* gene, is the major contributor to ALA content in seeds [18]. In soybean, while three *FAD3* genes contribute to ALA content in seeds [19], in flax [26] and perilla [5] it is controlled by two *FAD3* genes. In tree peonies, the expression levels of putative *FAD2* and *FAD3* were positively correlated with ALA content suggesting their role in synthesis of ALA via PC-derived pathway [1].

Recent transcriptomics of oil-rich tissues of oil palm [27], *Sacha inchi* [28], peanut [29], avocado [30], perilla [31], and various oilseeds [32], including tree peony seeds [1], led to the identification of key genes and regulators of TAG biosynthesis. The seed development process of tree peonies is complex, however, and the ALA accumulation is development- and species-dependent [1]. Here, we compared the developing seed transcriptome of *Paeonia rockii* and *Paeonia lutea*, a high and low ALA accumulating species, respectively. The RNA-seq data were analyzed to identify key metabolic steps that contribute to differential ALA accumulation. The data also led us to further clone *FAD2* and *FAD3* and characterize their role in ALA biosynthesis.

## 2. Results and Discussion

Among the developing seeds of nine wild tree peony species that were previously analyzed, seeds from *P. rockii* contained the highest FA content (271.82 mg/g seed DW), while *P. lutea* had the lowest (150.09 mg/g seed DW) [1]. In addition to ALA being the major contributor for the difference in FA content in the seeds of the two species, there is also phenotypic variation in terms of size and color (Figure 1). The seeds of *P. rockii* are significantly smaller and weigh less than those of *P. lutea* during mid and late stages of development (Figure 1C). Interestingly, while *P. rockii* seeds are pale yellow in color and darken during maturation (Figure 1A), seeds of *P. lutea* are very dark from the early stages itself (Figure 1B). Various studies on cultivated varieties of brassica, including quantitative trait loci mapping [33,34] and flax oil seeds [35], identified a strong negative correlation between dark seed coat (flavonoid content) and oil content. The correlation, if any, of lower FA and ALA content of *P. lutea* seeds, relative to *P. rockii* is associated with its dark seed color—and possible interaction between the lipid and flavonoid biosynthesis pathway—is yet to be determined. Here, the comparison of seed transcriptomes of both species during development was limited to identifying crucial steps and the characterization of key enzymes, including desaturases involved in ALA synthesis.

### 2.1. Transcriptome Data Reveals Lipid Pathways Associated with High ALA Content in Seeds

To identify key genes involved in ALA synthesis, three seed developmental stages of *P. rockii* and *P. lutea* (Figure 2B,C), with differential ALA accumulation (Figure 1C) were selected. Their transcriptome yielded over 271 million reads with 64,044 and 47,322 unigenes (https://www.ncbi.nlm.nih.gov/sra/SRP150148) for *P. lutea* and *P. rockii*, respectively (Appendix A; Appendix A). The transcriptomes were primarily annotated by BLASTX search against the NCBI nonredundant (Nr) protein database [36] and SwissProt (Appendix A) [37]. In BLASTX homology search with the cutoff E-value set at 10^−6^, 22,533 unigenes (35.2%) for *P. lutea* and 24,860 unigenes (52.5%) for *P. rockii* were identified. For both species, the three top-hit species were *Vitis vinifera*, *Juglans regia*, and *Theobroma cacao* (Appendix A). A total of 45,205 and 49,563 unigenes for *P. lutea* and *P. rockii*, respectively, were assigned at least one gene ontology (GO) terms. For both species, the most highly represented terms for the category of biological process were metabolic, cellular, and single-organism processes; catalytic activity and binding in the molecular function category; and cell, membrane, and organelle in the cellular component category (Appendix A).

Considering the interest in ALA and TAG biosynthesis in general, the lipid database with information for over 740 genes encoding proteins involved in acyl lipid metabolism (http://aralip.plantbiology.msu.edu/, accessed on November 2014) [38] was used to identify associated unigenes in tree peony transcriptome (Appendix A). More than 600 unigenes from each species were annotated as homologs to lipid metabolism genes and were categorized into 18 associated pathway groups by KEGG analysis (Appendix A). More than two-thirds of these unigenes in both species were represented in seven categories that primarily contribute to glycerolipid and FA metabolism (Figure 2A). The total transcript levels during seed development, expressed as reads per kilobase per million (RPKM), for five of these seven categories were significantly higher in *P. rockii* than in *P. lutea* (Figure 2A and Appendix A). Although some of the unigenes may overlap among the categories in KEGG analysis, it is noteworthy that the expression levels for genes involved in synthesis of FAs, particularly ALA and other unsaturated FAs, were significantly higher in the seed transcriptome of species that is richer in oil and specifically ALA content. Such a pattern of higher and differential expression levels for genes specifically encoding for enzymes involved in acyl lipid metabolism is consistent with transcriptome analyses of oil-rich seed and nonseed tissues [27,30,32].

Comprehensive transcriptome studies have previously demonstrated that the synthesis of TAG primarily involves at least 28 genes encoding for enzymes in FA biosynthesis in the plastid and TAG assembly in the ER [27,30,32], excluding oleosins, the packaging proteins (Appendix A). The total RPKM values of these unigenes in plastid and ER, annotated as homologs to TAG biosynthesis genes and expressed during seed development were compared between the two species (Figure 2B,C). Interestingly, despite the higher ALA content in *P. lutea* at 20 days, the total transcript levels in the plastid and ER remained similar between the species, which perhaps reflects the similar total oil content in seeds during their early developmental stages [1]. The ALA amount, however, was significantly higher in *P. rockii* compared to *P. lutea* during the later stages (60 and 80 days; Figure 1C). By 80 days, ALA content in *P. rockii* seeds was 2.4-fold higher than of *P. lutea* (*p* < 0.01; Figure 1C), which coincides with the significantly higher expression of FA synthesis genes in *P. rockii* than in *P. lutea* (*p* < 0.01; Figure 2B). In the ER, significant differences in seed transcript levels between the two species were noted only at 80 days (Figure 2C). Although the transcript levels may not always reflect enzyme activity, several studies have shown a tight correlation between the expression of select genes encoding for oil biosynthesis and oil content [27,30,32]. These data suggest that, in *P. rockii*, key genes involved in FA synthesis in plastid are upregulated during 60–80 days (Figure 1B) and TAG accumulation in the ER at 80 days (Figure 2C), the time point at which accumulation of ALA has peaked (Figure 1C).

### 2.2. Gene Expression for Select Enzymes in TAG Metabolism Is Associated with High ALA Content

Since there were significant differences between the two species in their ALA content and lipid gene expression in plastid during 60–80 days, transcript levels for 14 plastidial genes during that period of seed development were compared (Appendix A). Among the 14 plastidial FA synthesis genes, only six of them were significantly upregulated in *P. rockii* by at least 1.5-fold, relative to *P. lutea*, during 60–80 days developmental period (Figure 3A,B). Similarly, among 14 ER genes, expression for nine genes was more than1.5-fold higher in *P. rockii* relative to *P. lutea* at 80 days (Figure 4A,B), the time period at which ALA content and gene expression in ER between the species were significantly different (Figure 2C). A significantly higher expression for select genes in *P. rockii* than in *P. lutea* suggests that only a few genes in the plastid and ER, among the possible 28 genes are likely to contribute to the more active TAG metabolism and higher oil/ALA content in *P. rockii*.

A higher rate of FA accumulation in tree peonies occurs mostly during 20–60 days after pollination (DAP) of developing seeds [1]. In association, most core enzymes of FA synthesis in the plastid and TAG assembly in the ER showed higher transcript levels at 20 and 60 days relative to 80 days (Figure 5A,B), which coincides with seed maturation. Comparing the temporal expression profile of the six plastidial and nine ER genes that were significantly higher in *P. rockii* (Figure 3B and Figure 4B) during the period of oil accumulation revealed that the differences were mostly limited to two expression patterns: (1) the transcript levels for these upregulated genes in *P. rockii* were either several-fold higher, and/or (2) decreased at a lower rate, relative to *P. lutea*, during the highest oil accumulation period (Figure 5A,B). Among the FA synthesis genes in the plastid, the average transcript levels for pyruvate dehydrogenase complex (PDHC), β-ketoacyl-ACP synthase (KAS) II and III, and reductase (KAR), stearoyl-ACP desaturase (SAD) and fatty acyl thioesterase A (FATA) were significantly higher in *P. rockii* than *P. lutea* (Figure 3B and Figure 5A).

The expression of *PDHC* is associated with enzyme activity that drives the committed step in FA synthesis by generating acetyl-CoA. High expression levels for PDHC have been highly correlated with de novo FA synthesis and high oil content in seed and nonseed tissues. Particularly, accumulation of transcripts for E1β subunit of PDHC was strongly correlated with the formation of lipid within the developing embryo in *Arabidopsis* [39]; in *P. rockii*, its expression levels were 3.2-fold higher than in *P. lutea* at 80 days (Appendix A). Genes *KASIII* and *KAR* encode for enzymes that act on immediate downstream steps to acetyl-CoA synthesis in FA biosynthesis. Specifically, while KASIII is a rate-limiting enzyme catalyzes the initial condensation reaction between acetyl-CoA and malonyl-ACP to produce 3-ketoacyl-ACP, KAR subsequently reduces it to 3-hydroxyacyl-ACP. Overexpression of KASIII, although reduced the rate of FA synthesis, an increase in palmitic acid was noted in seeds of rapeseed [40] and jatropha [41]. In *P. rockii*, the expression levels for *KASII* were also high (Figure 5A), suggesting its synergistic role with *KASIII* in both enhancing the palmitic acid content and in its subsequent elongation to stearic acid. As such, overexpression of *JcKASII* in *Arabidopsis kasII* mutant increased 18-carbon FAs but decreased 16-carbon acyl chains in leaves and seeds [42].

Expression levels of *KASII*, *SAD*, and *FATA* are expected to affect FA chain length, saturation, and the relative proportion of 16:0 and 18:1 [43], respectively. The enzyme SAD is responsible for introducing the first double bond into 18:0-ACP to produce 18:1Δ9-ACP, an essential precursor for feeding the acyl editing process leading to ALA synthesis. Thus, while higher levels of *KASII* are expected to result in increased 18:0-ACP, *SAD* expression will result in its desaturation and *FATA* in termination and release of 18:1 free FA. Expression level for all the three genes is more than 4.5-fold higher in *P. rockii* than in *P. lutea* at 80 days (Appendix A), suggesting their strong influence in production of 18:1 as the preferred end product of plastidial FA biosynthesis. The temporal expression profile was also very similar between the three genes, with the highest expression levels noted at 60 days (Figure 5A). The higher expression levels of *SAD* have been correlated with 18:1 accumulation in avocado mesocarp [30]. Among the thioesterases A and B, FATA is known to prefer oleoyl-ACP as a substrate [44]. Higher transcript levels for FATA in *P. rockii* also suggests 18:1 as the main product of plastidial FA synthesis. These results together imply that at least six genes in the plastid might play a key role in higher oil content in *P. rockii*, and SAD could be the first key enzyme that drives ALA synthesis in tree peony seeds by providing the 18:1 precursor.

Although nine genes showed significant upregulation in TAG assembly in the ER at 80 days for *P. rockii*, relative to *P. lutea*, temporal profile (Figure 5B) suggests that desaturases and enzymes involved in acyl editing might be crucial in determining the final TAG content and composition as discussed below. The temporal profile for some of the unigenes related to Kennedy pathway (*GPDH*, *LPAAT*, *PAP*) and two phospholipases (PLC, PLD) was similar between *P. lutea* and *P. rockii* (Figure 5B). Curiously, the expression levels for *GPAT* were not only higher in *P. lutea* during the early stages of seed development but *DGAT* levels also remained higher and stable during 20 to 60 days, and further increased by 80 days (Figure 4B and Appendix A). These data might suggest that increase in TAG and ALA content in *P. rockii* is likely achieved through a route alternate to the Kennedy pathway.

### 2.3. Acyl Editing and Head Group Exchange Drives PUFA Synthesis in P. rockii

Membrane lipid, PC is a central intermediate in the flux of FAs and/or DAG substrates into TAG synthesis. The esterification of newly synthesized acyl-CoA to PC occurs via acyl-CoA:lyso-PC acyltransferase (LPCAT) [45]. Generally, the *sn*-1 position of PC is esterified with saturated or monounsaturated FA and *sn*-2 with PUFAs. The appropriate acyl composition of PC is further achieved after de novo synthesis and modification, which involves deacylation and reacylation cycles. At least 50% of the nascent PC undergoes deacylation of *sn*-2 position to produce lyso-PC by the action of PLA2 followed by a reacylation with the action of LPCAT. Free FAs released by the hydrolysis of phospholipids at the *sn*-2 position by PLA2 enter the acyl-CoA pool for TAG assembly (Figure 4A). Such acyl flux is crucial for the production of TAG containing high levels of PUFAs and thus both PLA2 and LPCAT serve as key enzymes in the process of converting PUFA-PC into PUFA-CoA [45,46,47].

The temporal expression profile of *PLA2* in *P. rockii* indicates that the levels are higher at early stages of seed development compared to the later time points but interestingly transcripts for PLA2 were not detected in *P. lutea* (Figure 5B). A specific PLA2, identified through transcriptome study of castor endosperm [32] was shown to play a significant role in acyl editing and thus contributing to hydroxy FA accumulation in castor [48]. Similarly, LPCAT enzymes from several plant species including *Arabidopsis* [49], *Ricinus communis* [50], and *Brassica napus* [51] were cloned and characterized and shown to be crucial for PUFA synthesis. In *N. benthamiana*, two LPCATs were shown to prefer C18:3 PUFA as an acyl donor [52]. Interestingly, the transcript levels of *LPCAT* were significantly higher in *P. lutea* than in *P. rockii*, especially during 60 and 80 days developmental stages (Figure 5B). All this suggests that *LPCAT* may not be a rate-limiting enzyme for ALA synthesis in tree peony, and more active acyl editing/remodeling pathway in *P. rockii* might allow for flux of C18:3 to enter the acyl-CoA pool or C18:3-PC for TAG assembly.

In addition to acyl editing, the majority of acyl flux through PC for TAG synthesis is controlled by the exchange phosphocholine headgroup between PC and DAG. There are at least three routes to generate DAG from C18-PC (Figure 4A) [45]. First, a lipase-based mechanism utilizing PLC, or PLD plus PAP; second, *via* PDCT which exchanges phosphocholine between DAG and PC [53]; and third, the reverse reaction of CPT [54]. While the transcript levels of *PLC*, *PLD*, and *PAP* did not differ significantly between the two species, it is noteworthy that the average transcript levels of *PDCT* and *CPT* were two-fold higher than in *P. lutea* (Figure 2A; Appendix A). Interestingly, in *P. rockii*, the RPKM for *PDCT* and *CPT* were three- to five-fold higher than the transcripts for *PLC*, *PLD*, and *PAP*. Furthermore, the temporal expression profile of both *PDCT* and *CPT* suggests higher activity during 20–60 days of seed development (Figure 5B), which coincides with high rate of oil biosynthesis (Figure 1C) [1]. Thus, unlike in *P. lutea*, *P. rockii* might have more than one effective PUFA-PC pathway to generate DAG necessary for TAG synthesis (Figure 4A). Significantly higher expression for *PDCT* and *CPT*—not only relative to *PLC*, *PLD*, and *PAP* but also *P. lutea* (Figure 4B and Figure 5B)—suggests that the exchange of phosphocholine between DAG and PC is preferably mediated by those two enzymes for C18 PUFA synthesis [12,55].

### 2.4. An Acyl-CoA Independent Pathway May Be a Preferred Terminal Step in ALA Accumulation

Acyltransferases DGAT and PDAT are two important rate limiting enzymes in TAG synthesis and regulating their expression can affect the ALA content in plants. Silencing of *DGAT1* in Camelina using artificial microRNA gave a substantial increase in ALA content in seeds [56]. In *Arabidopsis*
*dgat1* knockout, *PDAT1* expression was upregulated by 65% and there was also an increased ALA content in seeds from 18 to 40% compared to wildtype [57]. Interestingly, while overexpression of intrinsic *PDAT1* neither changed the FA profile nor lipid content in *Arabidopsis thaliana* [58], it is considered to mediate TAG synthesis and contribute up to 75% of wildtype oil levels in the absence of *DGAT1* [13]. Furthermore, the acyl composition of TAG was also highly enhanced with PUFAs at the *sn*-3 position [59]. Similarly, overexpression of *LuPDAT1* (*Linum usitatissimum* L.) in *Arabidopsis* also resulted in an enhanced ALA content [60], which supports that PDAT1 contributes to the transfer of ALA, from the *sn*-2 position of PC to the *sn*-3 position of DAG [57]. In *P. rockii*, the transcript level for *PDAT* was significantly higher than in *P. lutea*, and the converse was true for *DGAT* expression (Figure 4B and Figure 5B; Appendix A). The ALA accumulation pattern (Figure 1C) correlates with that of the *PDAT* expression, suggesting that an acyl-CoA independent pathway is the likely route for PUFA synthesis in tree peony seeds. The lower expression level of *DGAT* in *P. rockii* might drive more FAs into PC, which will subsequently aid in accumulation of more ALA into TAG. On the contrary, in *P. lutea*, the higher expression level of *DGAT* and the lower expression level of *PDAT* is likely to decrease the ALA accumulation in TAG. The higher expression of *Camelina sativa* PDAT1-A in seeds and ectopically in leaves also increased the ALA content by 45% compared with wildtype [61]. These data together suggest that *PDAT* expression levels and the ratio of *PDAT*/*DGAT* might determine the final ALA content in TAG.

### 2.5. Desaturases Play a Key Role in Driving the ALA Content in Seeds

The 18:1-ACP generated by SAD in the plastid is transported to ER as 18:1-CoA and incorporated into PC where it can be further desaturated to 18:3-PC by the action of FAD2 and FAD3, which introduce the second and third double bond into 18:1-PC, sequentially [62]. In *P. rockii*, more than 45% of the transcripts encoding for FA and TAG biosynthesis pathway proteins were mapped to SAD, FAD2, and FAD3 at 60 and 80 days (Appendix A). In addition, their transcript levels in both species increased from 20 to 60 days (Figure 5A,B), coinciding with their ALA accumulation pattern (Figure 1C). In *P. lutea*, however, the expression levels of all the three desaturases reduced faster in seeds from 60 to 80 days than in *P. rockii* (Figure 5B). The major differences in expression of *FAD2* and *FAD3* were mainly temporal suggesting that the desaturase genes were more effective with sustained expression in *P. rockii* than in *P. lutea*.

To further confirm the significance of the three desaturases associated with FA desaturation, their expression levels were evaluated in four stages of developing seeds using qRT-PCR. Both *FAD2* and *FAD3* levels in *P. rockii* were significantly higher than in *P. lutea* (Figure 6). Consistent with transcriptomic data (Figure 5B), these two genes were highly expressed in *P. rockii*, and lowly in *P. lutea* at 80 days (Figure 6). In 40 and 80 days seed, the *SAD* was highly expressed in *P. rockii* than in *P. lutea* (Figure 6). Both RNA-seq and qRT-PCR results showed that the transcripts of *FAD2* and *FAD3* were higher at 60 and 80 days in *P. rockii*, while in *P. lutea*, although they were higher at 60 days, expression reduced to very low levels at 80 days (Figure 5B and Figure 6). The temporal profile of *FAD2* and *FAD3* expression also agreed with the ALA accumulation pattern during the seed development of both peony species (Figure 1C) suggesting their control in ALA content. Similar transcriptome analysis of developing seed of perilla also revealed that the expression profile of *FAD3* gene is correlated with ALA synthesis in seed [31].

### 2.6. Molecular Cloning and Expression Analysis of PrFAD2 and PrFAD3

To investigate the role of critical desaturase genes in PUFA biosynthesis in ER, full-length coding DNA sequences of 1155 bp and 1308 bp, corresponding to putative *PrFAD2* (GenBank accession No. MG845869) and *PrFAD3* (GenBank accession No. MG845868), respectively, were identified in *P. rockii*. Their deduced amino acids shared a highly conserved 1-acyl-*sn*-glycerol-3-phosphate acyltransferase-related domain (Figure 7A and Figure 8A). Transmembrane domain prediction analysis revealed five and three transmembrane regions in deduced polypeptides encoded by *PrFAD2* and *PrFAD3* (Figure 7B and Figure 8B), respectively. Both FAD2 and FAD3 enzymes generally contain six [14,16] and four [17,63] transmembrane domains, respectively. Analysis of protein tertiary structure indicated that *Pr*FAD2 consisted of four α-helices (Figure 7C), while *Pr*FAD3 contained three α-helices (Figure 8C). Phylogenetic analyses of *Pr*FAD2 showed that it is closely associated with FAD2 proteins from *P. lactiflora* (AKE44629), *Eucalyptus grandis* (XP_010058087), *Citrus sinensis* (XP_006492862), and *C. clementine* (XP_006429873) (Figure 7D). On the other hand, *Pr*FAD3 showed 97.7%, 79.8%, and 74.1% similarity with the FAD3 homologs from *P. lactiflora* (AJA36814) and *V. vinifera* (*XP_002277573* and *XP_010650893*), respectively (Figure 8D).

To study tissue-specific expression of *PrFAD2* and *PrFAD3*, transcript levels in seeds at various developmental stages and other tissues in *P. rockii* were analyzed by qRT-PCR. Transcripts of both genes were detected in all the tissues tested; *PrFAD2* was constitutively expressed in floral and fruit tissues (calyxes, petals, stamens, pistils, and seeds) at a higher level than in leaves, stems, and roots, and stamens had a maximum transcript abundance (Figure 7E). A similar ubiquitous expression pattern for *FAD2* gene was observed in other plants such as in cotton and maize [64,65]. A marked increase in *PrFAD2* transcripts, however, occurred during seed development, with a peak at 80 days (S8) followed by a sharp decline (Figure 7F). The pistils and seeds exhibited significantly higher constitutive expression levels of *Pr*FAD3 than other tissues, with pistils and calyxes being highest and lowest, respectively (Figure 8E). Transcripts for *PrFAD3* rapidly and dramatically increased at different developmental stages of seeds, also reaching a peak at 80 days (S8) followed by a gradual reduction at 90 days (S9) and 100 days (S10) after pollination (Figure 8F). The expression pattern of *FAD3* of chia and perilla in different organs showed that while *FAD3* was expressed at varying levels in different organs, it was mostly expressed in the late stage of seed development, the period during which ALA was typically accumulated [66].

### 2.7. Overexpression of PrFAD2 and PrFAD3 Affects the ALA/LA Ratio in Arabidopsis Seeds

For a final validation of the biological functions of *Pr*FAD2 and *Pr*FAD3, transgenic *Arabidopsis* plants with constitutive overexpression of these two genes under the control of CaMV 35S promoter were generated. Three T_2_ homozygous lines with highest expression of *PrFAD2* (#17, #51, and #58) or *PrFAD3* (#10, #14, and #26), as determined *via* semi-quantitative RT-PCR, were selected for seed FA analysis (Figure 9A,C). Overexpression of *PrFAD2* resulted in a remarkable reduction in the ratio of ALA/LA (Figure 9A), but increased in transgenic lines overexpressing *PrFAD3*, compared to wildtype (Figure 9C). Furthermore, *PrFAD2*-overexpressing transgenic lines displayed elevated levels of C16:0, C18:0, and C18:2 but decreased levels of C18:1 and C20:0 (Figure 9B), while an obvious increase in C16:0, C18:1, C18:3, and C20:1 but decline in C18:2 and C20:0 levels were detected in *Arabidopsis* with *PrFAD3* overexpression (Figure 9D). These results demonstrate that *Pr*FAD2 and *Pr*FAD3 have opposing impact on ALA/LA ratio and alter FA composition of seed. *FAD2* gene from *V. labrusca* functionally complemented the *Arabidopsis*
*fad2* mutant [23]. In olive, higher expression of *FAD2* gene was associated with increased LA content [22], while the expression of *FAD2* from *Vernicia fordii* in yeast affected unsaturated FA metabolism [67]. Oil palm *FAD2* gene was found to only accept oleic acid as a substrate [68]. Conversely, in *Sacha inchi*, the expression level of *FAD3* gene in seeds was higher than in other organs and positively correlated with the higher percentage of ALA content in seed oil [28]. In addition, when *FAD3* gene from *Salvia hispanica* and *Perilla frutescens* was overexpressed in yeast and was grown in media supplemented with LA, transformed yeast utilized LA and produced a significant amount of ALA compared to wildtype yeast [66]. Together with the various observations made in plants, the role of peony *FAD2* and *FAD3* genes in LA and ALA acid synthesis, respectively can be ascertained unequivocally.

## 3. Materials and Methods

### 3.1. Plant Material

Among the two tree peony species used in this study, *P. rockii* grows about 0.8 to 2 m in height with 15–21 leaflets. The flowers are white, pink or red with purple patches on the bottom of the petals, and fruits are composed of five carpels (Figure 1A). *P. rockii* is mainly distributed in Shaanxi, Gansu, and Hubei, China. Plants of *P. lutea* grow 0.5–1.6 m in height and bear nine leaflets; flowers are yellow, the number of carpels is 3–6 (Figure 1B); mainly distributed in Yunnan, Xizang, Sichuan, China. Seeds from *P. rockii*, and *P. lutea* varied in their size and color during seed development (Figure 1). Among them, the seed color is darker and the seed is larger in *P. lutea* (Figure 1). 

Seeds of *P. rockii* and *P. lutea* were collected in 2015 at the Wild Tree Peony Germplasm Repository, Gansu Forestry Science and Technology Extension Station, China (36°03′ N, 103°40′ E). Wild tree peony seeds were introduced to the germplasm repository and were cultivated under same environmental and cultivation conditions for 13 years. We monitored their seed development process from pollination until maturation from May to August 2015. Pods from two wild species were collected all the developmental stages (S1 to S10) from 0 to 100 days at 10 days intervals. The ALA content of two wild species seeds was measured by GC-MS as previously described [1]. For the transcriptome sampling, seeds of *P. rockii* and *P. lutea* at three developing stages (20, 60, 80 days) were collected from the same individual, and pods with the same developing stage from three plants comprised of two replicates. For qRT-PCR, seeds from ten developing stages (S1 to S10) and other tissues (root, stem, leaf, calyx, petal, stamen, and pistil) of *P. rockii* were collected from three plants comprised of three replicates. All samples were flash frozen in liquid nitrogen and stored at −80 °C.

### 3.2. RNA Extraction, Sequencing and De Novo Assembly

Total RNA was extracted from seeds of *P. rockii* and *P. lutea* collected at 20, 60 and 80 days separately using the TIANGEN RNA Prep Pure Plant kit (Tiangen Biotech Co. Ltd., Beijing, China) following the manufacturer’s instructions, and mRNA was enriched by Oligo (dT) magnetic beads. RNA integrity was evaluated with an RNase free agarose gel electrophoresis, and the yield and quality of the RNA samples were determined using a 2100 Bioanalyzer (Agilent Technologies, Santa Clara, CA). The cDNA library was constructed by using a NEBNext^®^ UltraTM RNA Library Prep Kit for Illumina (NEB, Ipswich, MA, USA), and it was sequenced using an Illumina HiSeq^TM^ 4000 platform (Illumina Inc., San Diego, CA, USA) by Gene Denovo Biotechnology Co. (Guangzhou, China). Before assembly, adapter sequences, more than 10% of unknown nucleotides (N), and low-quality reads containing more than 40% of low quality (*Q*-value ≤ 10) bases were removed from the raw reads to gain more reliable results. After that, the high-quality clean reads from all samples of the same wild species were merged together and assembled using Trinity package [69] to construct unique consensus sequences as the reference sequences.

### 3.3. Normalization of Gene Expression Levels and Identification of Differentially Expressed Genes

The SOAP aligner v.2.21 tool (http://soap.genomics.org.cn/soapaligner.html#down2, accessed on May 2010) was used to align sequencing reads to the reference sequences. To eliminate the influence of different sequencing discrepancies and gene lengths on the gene expression calculation, the expression level of each gene was measured as RPKM (Reads Per kb per Million reads) based on the number of uniquely mapped reads. The longest transcript was selected to calculate the RPKM for genes with more than one alternative transcript.

To infer the transcriptional changes during seed development in the two tree peony species, differentially expressed genes (DEGs) after 60 and 80 days of pollination were identified by comparing the expression levels at 60 days with those at 20 days and the level at 80 days with those at 60 days in *P. rockii* and *P. lutea*, respectively. The false discovery rate (FDR) was calculated to adjust the threshold of *p*-value [70] to correct for multiple testing. Transcripts with a fold change ≥ 2 and an FDR < 0.05 in a comparison as differentially expressed between the two time points [71].

### 3.4. Functional Annotation, GO and KEGG Classification

All expressed genes were functional annotated against four databases, including NCBI nonredundant protein database (NR), Kyoto Encyclopedia of Genes and Genomes (KEGG), Universal Protein (UniProt), and Eukaryotic Orthologous Groups (KOG), by BLASTX searches (E-value < 10^−5^) in Blast2GO [72]. For the gene matched to multiple protein sequences, the protein with the highest similarity score was considered as the optimal annotation.

GO functional classification of the DEGs was implemented by the WEGO [73], in order to demonstrate the distribution difference of gene functions between the two species into the three main ontology categories: molecular function, cellular component and biological process. Over-presented GO terms were identified using a hypergeometric test with a significance threshold of 0.05 after the Benjamini and Hochberg FDR correction. KEGG enrichment analysis of the DEGs was performed using the KOBAS 2.0 software [74].

### 3.5. Quantitative Real-Time PCR

Quantitative real-time PCR (qRT-PCR) was performed by using a SYBR^®^ Premix Ex Taq™ kit (DRR041A, Takara, Dalian, China) in a StepOne™ Real-Time PCR System (Applied Biosystems, Foster City, CA). The qRT-PCR reaction systems were as follows: 95 °C for 15 s, followed by 40 cycles of 95 °C for 5 s, 58 °C for 30 s, and 72 °C for 31 s. The fluorescence data were collected and analyzed with StepOne™ analysis software during the 72 °C step. 18S–26S internal transcribed spacer (ITS) was used as a reference gene to normalize the expression data, and the 2^−^^△△*C*t^ values were shown as relative expression levels [75]. *AtActin7* was used as an endogenous control gene to normalize the semi-quantitative RT-PCR analysis of *PrFADs* in the wildtype and 35S::*PrFADs* plants. All primers used for the assessment of transcript levels were listed in Appendix A. 

### 3.6. Cloning and in Silico Analysis of Desaturases

Based on transcriptome data, the full-length open reading frames of putative *PrFAD2* and *PrFAD3* were identified, PCR-amplified from 40 days seeds of *P. rockii* using specific primers (Appendix A) and cloned into pUCm-T vector (SK2212, Sangon, Shanghai, China) for sequencing and expression into binary vector. Blast (http://blast.ncbi.nlm.nih.gov, accessed on April 2010) was used for homology search of sequences at nucleotide and protein levels. TMHMM2.0 (http://www.cbs.dtu.dk/services/, accessed on October 2011) was used for transmembrane prediction analysis of *PrFAD2* and *PrFAD3*. Red lines indicate the transmembrane, blue lines indicate the inside, pink lines indicate the outside. The tertiary structure prediction was conducted using Phyre Version 0.2 (Imperial College London, South Kensington Campus, London SW72AZ, UK) [76]. Based on multiple sequence alignment by MUSCLE (http://www.ebi.ac.uk/Tools/msa/muscle/, accessed on June 2013), and the neighbor-joining method with MEGA software (version 5.1, Tokyo Metropolitan University, Hachioji, Tokyo, Japan) [77] was used to construct the phylogenic tree. Boot-strap values as a percentage of 1000 replicates are indicated at corresponding branch nodes. Scale bar represents the number of amino acid substitutions per site. 

### 3.7. Overexpression and Stable Transformation

Both *PrFAD2* and *PrFAD3* were ligated into the KpnI-PstI and SalI-PstI restriction sites, respectively of binary vector pCAMBIA1300 under the control of the CaMV 35S promoter. The generated 35S::*PrFAD2* and 35S::*PrFAD3* constructs were transformed into wildtype *Arabidopsis* using the floral dip method [78]. The harvested seeds were planted on 1/2 MS plates containing 20 mg/L hygromycin and placed in plant growth chamber (RXZ, Jiangnan Instrument, Ningbo, China) at 22 ± 2 °C under a 14/10 h light/dark (120 μmol·m^−2^·s^−1^) cycle. Hygromycin-resistant seedlings with well-established roots and green leaves were selected as *T*_0_ transformants, and transferred to moistened potting soil from the 1/2 MS plates. The positive transformants were confirmed by PCR. Independent transgenic lines exhibiting a 3:1 segregation of 20 mg/L hygromycin resistance in the *T*_1_ generation were selected. Homozygous *T*_2_ transgenic lines showing 100% survival on hygromycin-containing medium were finally established. Mature *Arabidopsis* seeds were collected from *T*_2_ homozygous lines for further analysis. All experimental plants (including controls) were grown at the same time in the same location. The FA measurement of homozygous transgenic lines seeds was measured by GC-MS as described previously [79].

### 3.8. Statistical Analysis

All experiments included three biological replicates and technical replicates as previously indicated. Mean ± standard deviation (SD) was determined, and unless otherwise noted, one-way analysis of variance was carried out using SPSS (version 17.0 for Windows; SPSS Inc, Chicago, IL, USA). Unless otherwise noted, asterisk (*) indicates significant differences at *p* < 0.05 level of least significant difference (LSD) test between two species or test groups; different small letters indicate significant differences at *p* < 0.05 level of LSD test between two species at the same time point; different capital letters indicate significant differences at *p* < 0.05 level among different time points in the same genotype.

## 4. Conclusions

In plants, the final TAG content and composition is majorly influenced by the end product of plastidial FA biosynthesis, the acyl editing through PC-derived pathway and their assembly in the ER. In this study, we identified that six key plastidial genes, *PDHC*, *KASIII*, *KAR*, *KASII*, *SAD*, and *FATA* are likely driving the generation of 18:1 acyl group as the major product that feeds into ALA synthesis in the ER. In peony seeds, acyl editing and PDAT-mediated TAG assembly are likely to play a key role in subsequent ALA accumulation. Both FAD2 and FAD3 are the two most important desaturase enzymes that contribute to PUFA synthesis through acyl editing in PC-derived pathway in the ER. The regulation of FAD2 and FAD3 determines the final seed oil composition, especially the ratio of LA to ALA. By cloning *FAD2* and *FAD3* from *P. rockii* and their overexpression in *Arabidopsis*, we showed the role of these two desaturases in modulating the ALA/LA ratio in seeds. In addition to affecting PUFA composition, overexpression of *PrFAD2* and *PrFAD3* in *Arabidopsis* significantly increased the LA and ALA production, respectively (Figure 9). These results indicate that FAD2 and FAD3 via acyl editing mechanisms control the level of ALA in tree peony seeds.

Overall, our study concludes that the high ALA content in *P. rockii*, relative to *P. lutea* (Figure 10), is result of four major steps in the production of ALA. First, plastid FA synthesis seems to serve as a source by primarily providing C18:1 for further incorporation into PC in the ER. Second, the acyl editing pathway allows for desaturation of C18:1-PC by FAD2 and FAD3 sequentially to produce C18:3-PC. Third, ALA from PC is incorporated into DAG by different exchange routes including *PLA2*, *PDCT*, and *CPT*. Finally, *DGAT* and *PDAT* might function in an opposing manner controlling the final step in TAG assembly. We conclude that, along with the role of desaturases, modulation of lower *DGAT* and higher *PDAT* expression levels are expected to enhance ALA content of TAG in tree peony seeds. Further characterization of these key metabolic steps and their regulation by genetic manipulation are expected to generate tools to increase the ALA production not only in tree peony seeds but also in other oilseeds of agronomic interest.

## Figures and Tables

**Figure 1 ijms-20-00065-f001:**
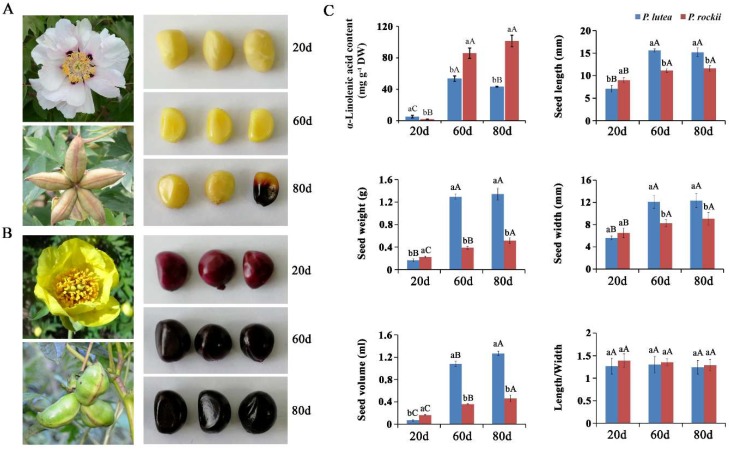
Phenotypic variation between the two tree peony species. (**A**) The flower, fruit and developing seeds of *P. rockii* and (**B**) *P. lutea;* (**C**) seed variation analysis. Different small letters indicate significant differences between the species at the same time point at *p* < 0.05 level, as determined by the least significant differences test. Different capital letters indicate significant differences among different time points for the same species.

**Figure 2 ijms-20-00065-f002:**
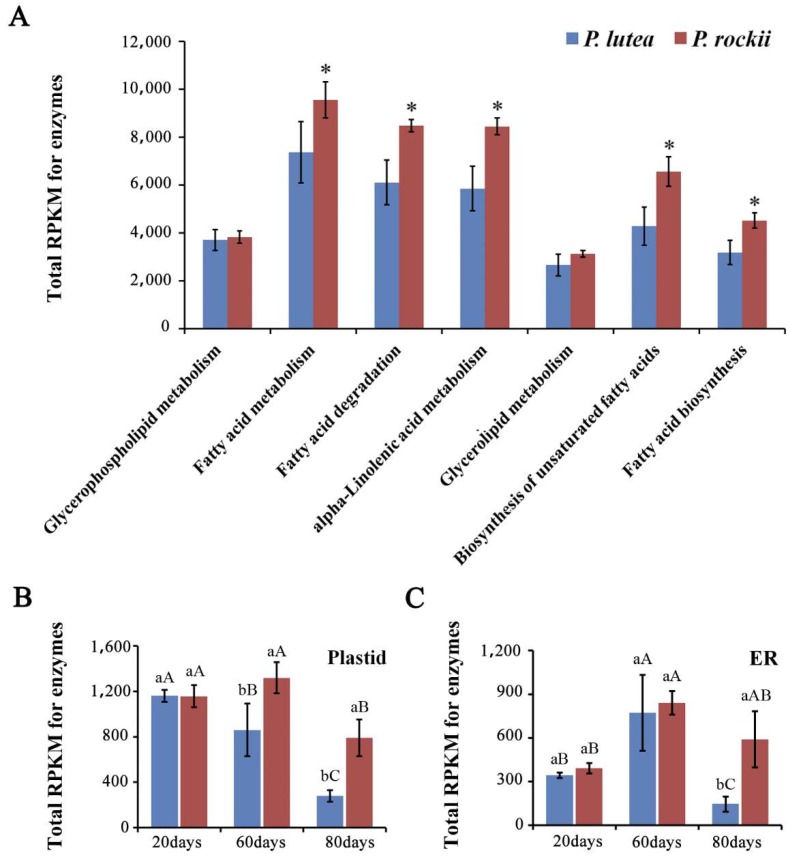
Transcriptome analyses of developing seeds of *P. lutea* and *P. rockii*. Total transcript levels for unigenes representing the (**A**) top seven lipid pathways (Appendix A), (**B**) fatty acid synthesis in the plastid and (**C**) triacylglycerol assembly in the endoplasmic reticulum (ER; Appendix A). RPKM: reads per kilobase per million. The asterisks indicate significant differences (* *p* < 0.05, Student’s *t*-test). Different small letters indicate significant differences between the species at the same time point at *p* < 0.05 level, as determined by the least significant differences test. Different capital letters indicate significant differences among different time points for the same species.

**Figure 3 ijms-20-00065-f003:**
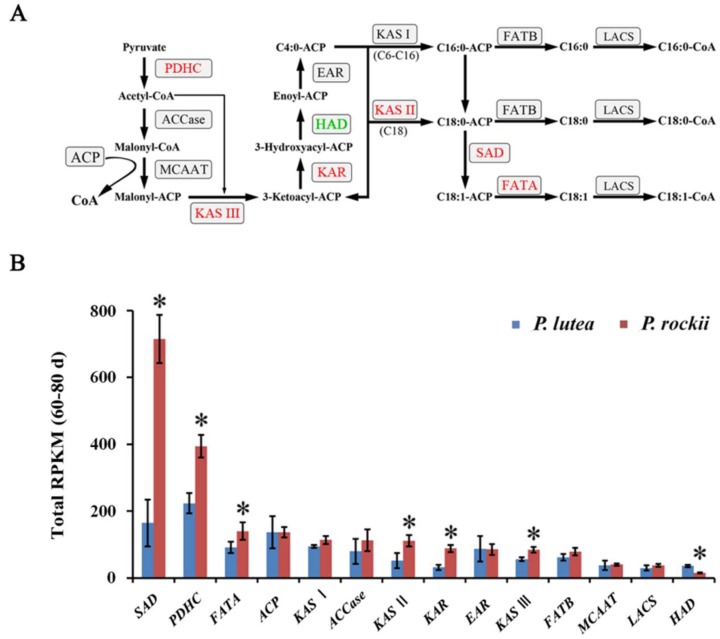
Differential lipid gene expression in plastid between the two species. (**A**) Schematic outline of the genes encoding for fatty acid biosynthesis in plastid and (**B**) their total RPKM during 60 to 80 days of developing seeds. For gene abbreviations see Appendix A. Genes represented in red letters indicate significant difference in their expression levels between the two species, with a ratio of *P. rockii* to *P. lutea* ≥1.5, and genes represented in green letters indicate the *P. rockii* is significantly lower than *P. lutea* in their expression levels. RPKM: reads per kilobase per million. The asterisks indicate significant differences (* *p* < 0.05, Student’s *t*-test).

**Figure 4 ijms-20-00065-f004:**
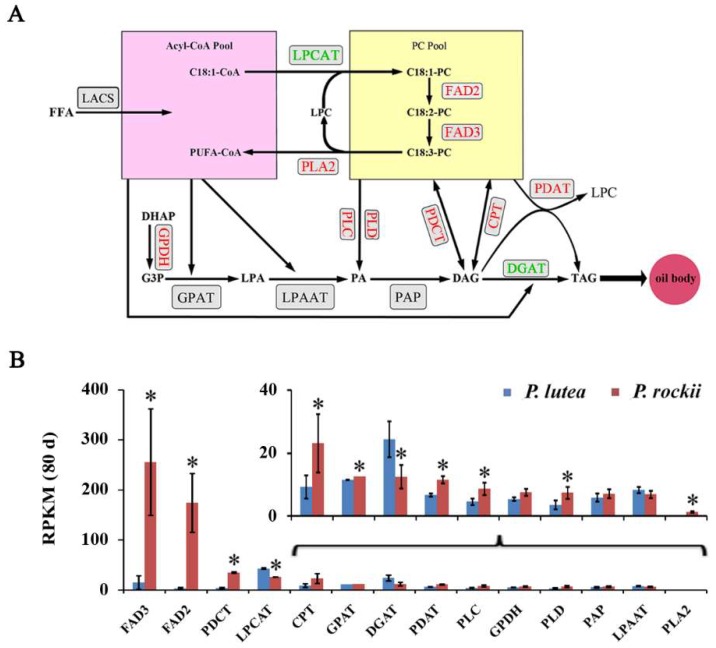
Differential lipid gene expression in endoplasmic reticulum (ER) between the two species. (**A**) Schematic outline of genes encoding for triacylglycerol assembly in ER and their (**B**) transcript levels in matured seeds (80 days). For gene abbreviations see Appendix A. Genes represented in red letters indicate significant difference in their expression levels between the two species, with a ratio of *P. rockii* to *P. lutea* ≥1.5; and genes represented in green letters indicate the *P. rockii* is significantly lower than *P. lutea* in their expression levels. RPKM: reads per kilobase per million. The asterisks indicate significant differences (* *p* < 0.05, Student’s *t*-test).

**Figure 5 ijms-20-00065-f005:**
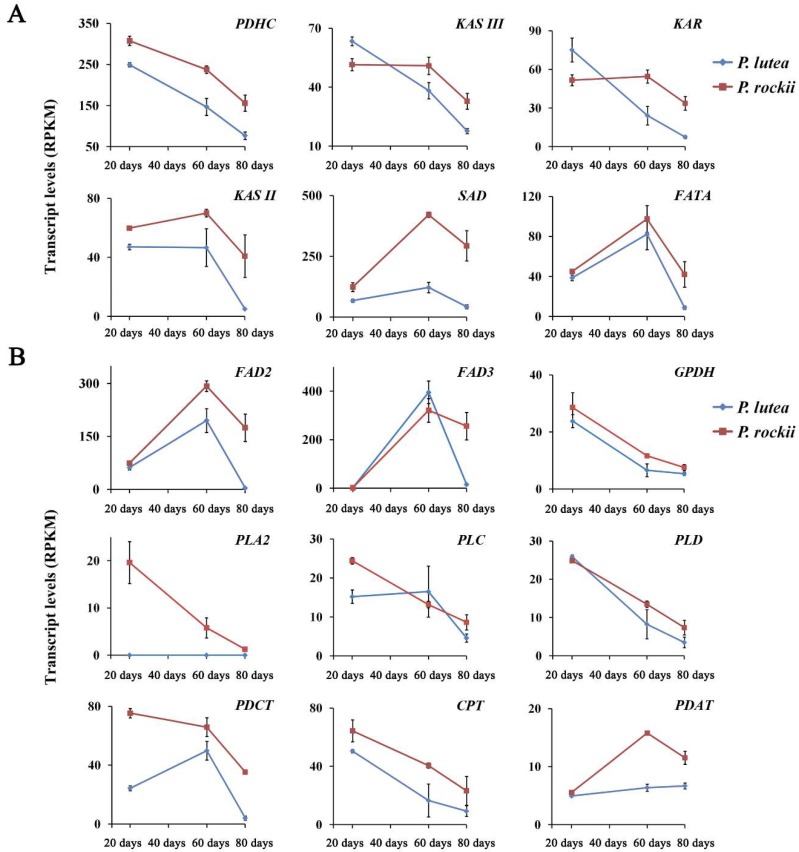
Comparison of expression pattern for select genes upregulated in *P. rockii* with *P. lutea* in (**A**) plastid and (**B**) endoplasmic reticulum.

**Figure 6 ijms-20-00065-f006:**
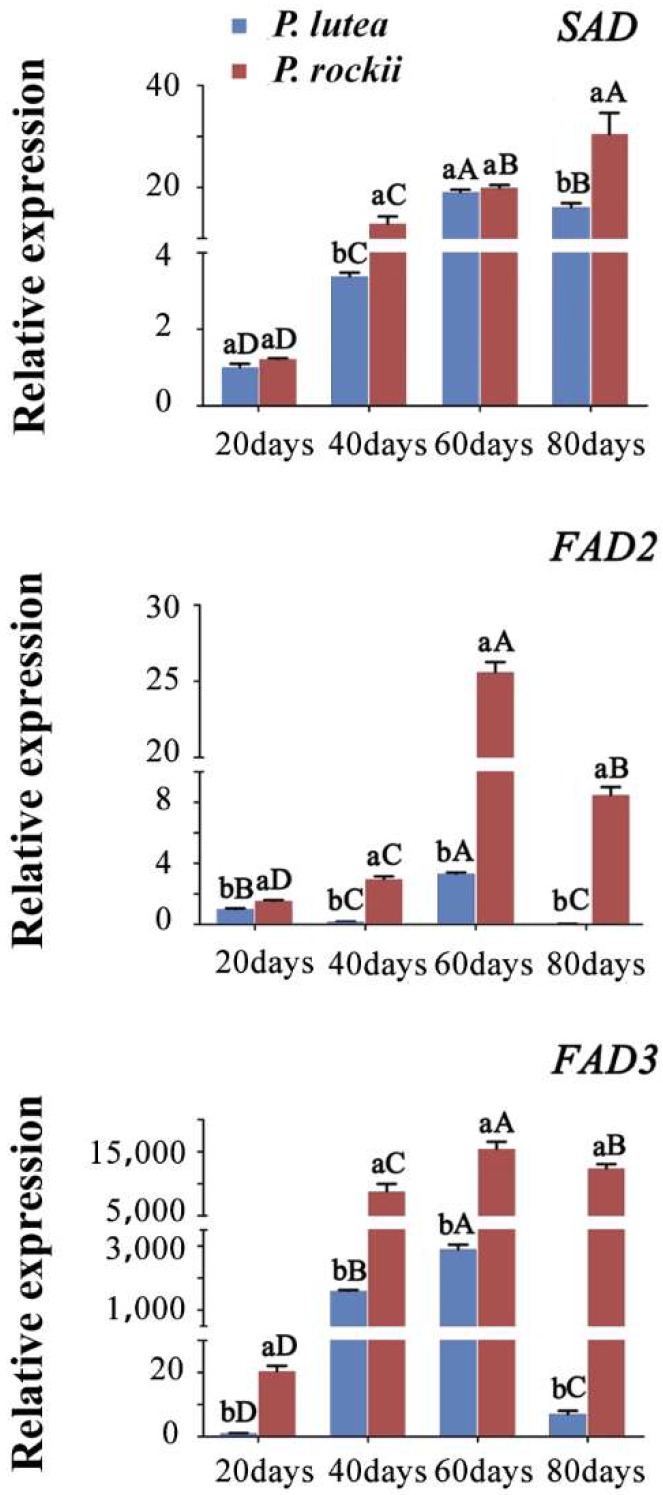
Validation of expression pattern for desaturases by qPCR. For gene abbreviations see Appendix A. Different small letters indicate significant differences between the species at the same time point at *p* < 0.05 level, as determined by the least significant differences test. Different capital letters indicate significant differences among different time points for the same species.

**Figure 7 ijms-20-00065-f007:**
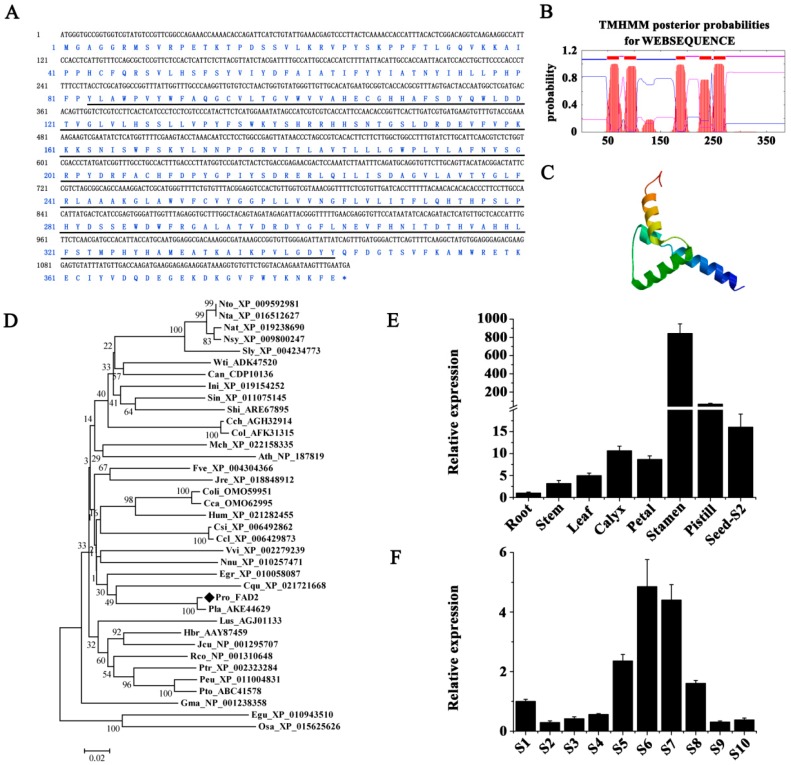
Identification, characterization, and expression analysis of *PrFAD2* in *P. rockii*. (**A**) Nucleotide and deduced amino acid sequence of *PrFAD2*. Solid lines indicate the conserved 1-acyl-*sn*-glycerol-3-phosphate acyltransferase-related domain. Prediction of the (**B**) transmembrane domains and (**C**) tertiary structure of *Pr*FAD2 protein. (**D**) Phylogenetic comparison of the FAD2 proteins in plants (refer to Appendix A for complete names of the plant species). (**E**) Tissue-specific expression of *PrFAD2*. (**F**) *PrFAD2* expression during *P. rockii* seed development. S1 to S10 indicate ten time points from 0 to 100 days at 10 days intervals. 18S–26S ITS was used as an internal control.

**Figure 8 ijms-20-00065-f008:**
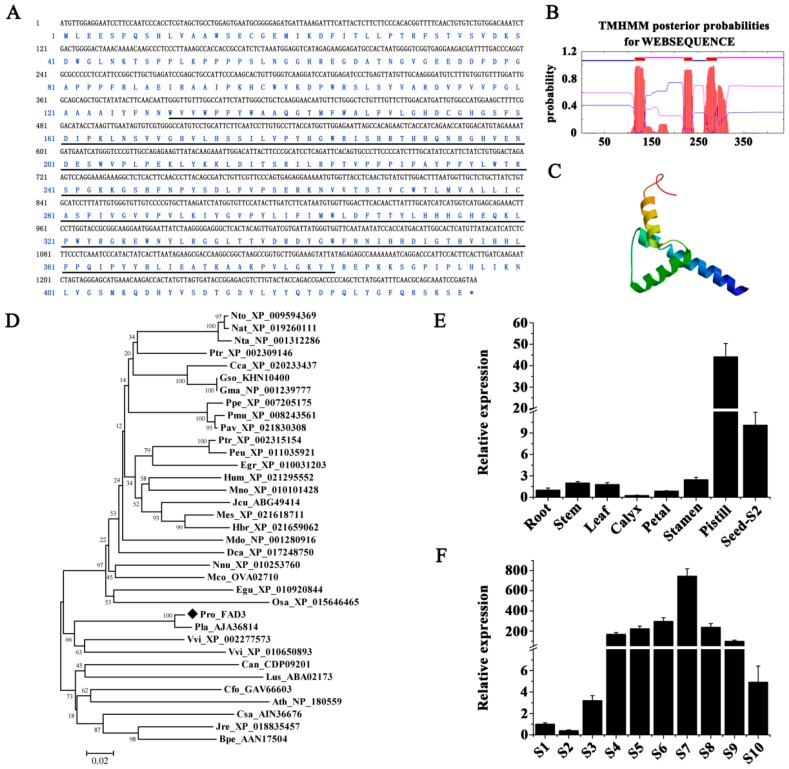
Identification, characterization, and expression analysis of *PrFAD3* in *P. rockii*. (**A**) Nucleotide and deduced amino acid sequence of *PrFAD3*. Solid lines indicate the conserved 1-acyl-*sn*-glycerol-3-phosphate acyltransferase-related domain. Prediction of the (**B**) transmembrane domains and (**C**) tertiary structure of *Pr*FAD3 protein. (**D**) Phylogenetic comparison of the FAD3 proteins in plants (refer to Appendix A for complete names of the plant species). (**E**) Tissue-specific expression of *PrFAD3*. (**F**) *PrFAD3* expression during *P. rockii* seed development. S1 to S10 indicate ten time points from 0 to 100 days at 10 days intervals. 18S–26S ITS was used as an internal control.

**Figure 9 ijms-20-00065-f009:**
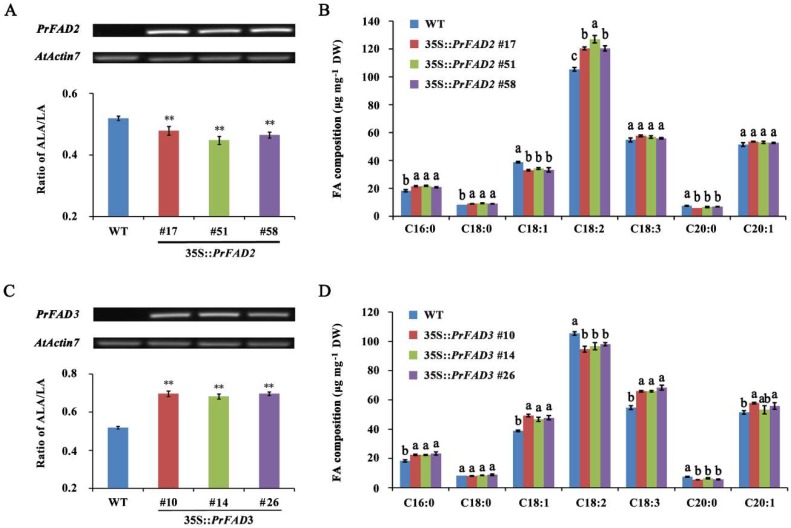
Overexpression of *PrFAD2* and *PrFAD3* in the wild-type *Arabidopsis* seeds. (**A**) Semi-quantitative RT-PCR and ALA/LA ratio, and (**B**) FA composition analyses of 14 days wildtype and 35S::*PrFAD2* (#17, #51 and #58) seeds. (**C**) Semi-quantitative RT-PCR and ALA/LA ratio, and (**D**) FA composition analyses of 14 days wildtype and 35S::*PrFAD3* (#10, #14, and #26) seeds. *AtActin7* was used as an endogenous control for semi-quantitative RT-PCR. The asterisks indicate significant differences (** *p* < 0.01; Student’s *t*-test). Different small letters indicate significant differences at *p* < 0.05 level.

**Figure 10 ijms-20-00065-f010:**
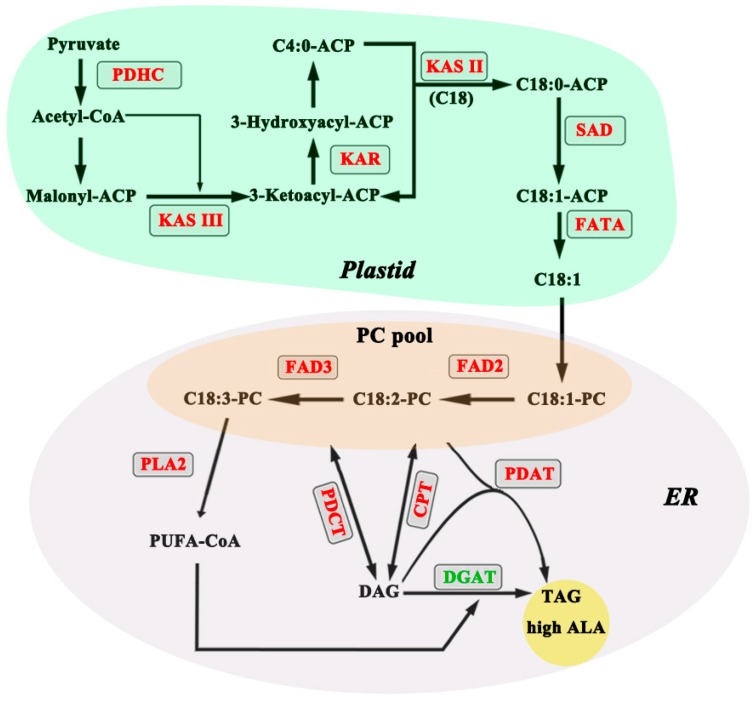
Schematic diagram of high α-linolenic acid (ALA) production. Genes represented in red letters indicate significant difference in expression levels between the two species, with a ratio of *P. rockii* to *P. lutea* ≥ 1.5; and genes represented in green letters indicate the *P. rockii* is significantly lower than *P. lutea* in expression levels. For abbreviations see Appendix A.

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
