# Peer review of "Comparative Transcriptome Analysis Reveals an Efficient Mechanism for α-Linolenic Acid Synthesis in Tree Peony Seeds"

_ijms, 2018, doi:10.3390/ijms20010065_

Round 1
Reviewer 1 Report
Manuscript (ijms-400077) “Comparative transcriptome analysis reveals the role of fatty acid desaturases in preferential synthesis of α-linolenic acid in tree peony seeds” by Zhang et al. presents an interesting study about the high-throughput sequencing of the Tree peony transcriptome in seed.
However despite the large amount of work done and data presented, the methodology and results presents some inconsistencies and deficiencies. Mainly, authors must add more genes in the RTqPCR validation. Only three genes is not enough.
For these reasons, this manuscript is acceptable for publication in International Journal of Molecular Sciences AFTER A MODERATE/MAJOR REVISION.
The major points for the revision of the manuscript are:
In the Abstract and Introduction, the scientific name of tree peony must be incorporated (genus and species).
Objective of the work must be clearly specified in a separated paragraph excluding bibliography.
Plant material assayed must be better explained. In my opinion a clear phenotypic characterization is necessary of the two assayed species. By one side, in the Methodology sections section authors must include the phenotypic evaluation protocol. By other side, a new Table of Figure is required with the completed phenotypic evaluation. Authors must separate this information from the Figure 1. In addition, in the Results and Discussion section authors must include the phenotypic evaluation results as a separated section.
A new Figure with the fatty acid synthesis in the plastid should be incorporated.
RNA-Seq data and qPCR data from Figure 4 should be statistically compared. Author must incorporate a correlation coefficient of data between RNA-Seq and qPCR. In addition, I higher number of genes must be incorporated. Only three genes is not enough.
Figure 5 should be spliced in other two figures to increase its quality.
Author Response
Point 1: In the Abstract and Introduction, the scientific name of tree peony must be incorporated (genus and species).
Response 1: Included in the abstract. The first sentence in the introduction includes the description of Genus, Species and Family; introducing the name again would be redundant. We use the full scientific name when we first mentioned Paeonia rockii and Paeonia lutea (line 84).
Point 2: Objective of the work must be clearly specified in a separated paragraph excluding bibliography.
Response 2: Objective of the work is described in separate paragraph (lines 80-87).
Point 3: Plant material assayed must be better explained. In my opinion a clear phenotypic characterization is necessary of the two assayed species. By one side, in the Methodology sections section authors must include the phenotypic evaluation protocol. By other side, a new Table of Figure is required with the completed phenotypic evaluation. Authors must separate this information from the Figure 1. In addition, in the Results and Discussion section authors must include the phenotypic evaluation results as a separated section.
Response 3: To highlight the differences in the seed phenotype for two Paeonia species we added additional images in Figure 1 and an improved description of the phenotype in methods section as well as in the results section as suggested. As such we moved RNA-seq data from Figure 1 to a separate figure (new Figure 2).
Point 4: A new Figure with the fatty acid synthesis in the plastid should be incorporated.
Response 4: Since we already have pathway for plastid and ER in Figure 2 but not clearly visible, we split that figure into two separate figures to enhance the visualization. Old Figure 2 is now Figure 3 & 4.
Point 5: RNA-Seq data and qPCR data from Figure 4 should be statistically compared. Author must incorporate a correlation coefficient of data between RNA-Seq and qPCR. In addition, I higher number of genes must be incorporated. Only three genes is not enough.
Response 5: We had data for more genes that were not included previously. As per the suggestion, a comparison between RNA-seq and qRT-PCR data are presented for ten select genes (Supporting Figure S4) along with correlation analyses (Supporting Figure S5), which showed R2 > 84%.
Point 6: Figure 5 should be spliced in other two figures to increase its quality.
Response 6: We prefer the format of both previous Figure 5 & 6 as we presented since they make it easier to compare. These figures are currently Figure 7 & 8.
Reviewer 2 Report
The paper entitled "Comparative transcriptome analysis reveals the role of fatty acid desaturases in preferential synthesis of a-linolenic acid in tree peony seeds” by Qingyu Zhang et al. reported that the expression levels of genes involved in plastidial fatty acid synthesis, acyl editing, desaturation, and triacylglycerol assmbly in the ER are revealed to be responsible for high and low ALA contents in seeds by the comparison of expression level of acyl lipid metabolism encoding genes in seeds between P.rockii and P.lutea, which accumulate high and low ALA in developing seeds, respectively,
In general, it is useful data in this field and the paper is well written.
I have several minor points and questions.
Figure 1. A, the pictures need scales. P.lutea looks larger than P.rockii. If possible, the data of total oil contents per a seeds (both P.rockii and Plutea) should be presented. Furthermore, the color of seed coat is different between these seeds. Is it related to oil contents or composition of fatty acids? In case of Brassica napus, seeds with pale color seed coart contain more oil than one with dark color seed coat.
Figure 7 A and D. the position of WT lane should be adjusted (right end or left end).
I think total TAG or fatty acid contents of both WT and transgenic plants should be presented. We want to know whether the overexpression of PrFAD2 or PrFAD3 affect the total oil contents in seeds.
Several letters or sentences are in different letter size or different letter style (p3, L96; p4, L154-157; p8, L278-280.
Author Response
Point 1: Figure 1. A, the pictures need scales. P. lutea looks larger than P. rockii. If possible, the data of total oil contents per a seeds (both P. rockii and P. lutea) should be presented. Furthermore, the color of seed coat is different between these seeds. Is it related to oil contents or composition of fatty acids? In case of Brassica napus, seeds with pale color seed coart contain more oil than one with dark color seed coat.
Response 1: We published the total fatty acid content of 9 wild species (Front. Plant Sci. 2018, 9, 106, DOI: 10.3389/fpls.2018.00106) and the results showed that the total fatty acid content of P. rockii was highest (271.82 mg/g seed DW), and P. lutea was lowest (150.09 mg/g seed DW), and we also found that α-linolenic acid was the major fatty acid in P. rockii. This is the main reason for our follow-up study. We thought presenting the total fatty acid data in this paper would be redundant but to refresh our readers to this extent, we added a sentence in our results (line 89-91).
We added phenotypic data for seeds from P. rockii, and P. lutea to the current figure that provides additional clarification on size and color. While there are studies that show correlation between seed color and oil content, such evaluation for tree peonies is yet to be made. We added a comment and three new references to this extent in the beginning of our results section (lines 89-103).
Point 2: Figure 7 A and D. the position of WT lane should be adjusted (right end or left end).
Response 2: We have modified it and also made few additional changes to make the figures compact and clear and reduce redundancy. The old Figure 7 is now Figure 9.
Point 3: I think total TAG or fatty acid contents of both WT and transgenic plants should be presented. We want to know whether the overexpression of PrFAD2 or PrFAD3 affect the total oil contents in seeds.
Response 3: We have measured the total fatty acid content of transgenic plants, which show moderate increase for overexpression of both FAD2 and FAD3. But our sample size was small and we would like to work with additional lines before we could present the data.
Point 4: Several letters or sentences are in different letter size or different letter style (p3, L96; p4, L154-157; p8, L278-280.
Response 4: Seems there was some error after converting to PDF; we will make sure that this will not be an issue in final version.
Reviewer 3 Report
Comments
In this manuscript authors compared the developing seed transcriptome of P. rockii and P. lutea. They identify key metabolic steps that contribute to differential ALA accumulation. Also clone FAD2 and FAD3, and characterize their role in ALA biosynthesis. There are, however, some important concerns that need to be resolved before the publication of this paper:
1. Tittle of this manuscript is much longer and should be reduced.
2. Authors selected 14 plastidial genes for comparison during the period of seed development. What is basis of selection of these 14 genes. Are they randomly selected?
3. Discussion section is not connected to results. Since, author mostly described their results. So, I suggested that authors should improve the discussion section. They have to discuss their results and comparison with some earlier and recent published papers in more depth.
4. Is the RNA-seq data is submitted in any public domain. If submitted please provide accession IDs.
Author Response
Point 1: Tittle of this manuscript is much longer and should be reduced.
Response 1: Edited to “Comparative transcriptome analysis reveals an efficient mechanism of α-linolenic acid synthesis in tree peony seeds”.
Point 2: Authors selected 14 plastidial genes for comparison during the period of seed development. What is basis of selection of these 14 genes. Are they randomly selected?
Response 2: Lipid database with information for over 740 genes encoding proteins involved in acyl lipid metabolism (http://aralip.plantbiology.msu.edu/) was used to identify associated unigenes, more than 600 unigenes from each genotype were annotated as homologs to lipid metabolism genes (Supporting Table S2). Several previous publications on oilseed crops have identified that synthesis of TAG primarily involves at least 28 genes encoding for enzymes in FA biosynthesis in the plastid and TAG assembly in the ER, excluding oleosins, the packaging proteins (Supporting Table S3 and S4). We selected 14 plastidial genes and they encode for all the major steps in fatty acid synthesis in plastid. Explanation to this extent is included in the manuscript in the section “2.1. Transcriptome data reveals lipid pathways associated with high ALA content in seeds”.
Point 3: Discussion section is not connected to results. Since, author mostly described their results. So, I suggested that authors should improve the discussion section. They have to discuss their results and comparison with some earlier and recent published papers in more depth.
Response 3: We reworked on the results sections to add additional depth to our discussion. Since we also had a conclusion section, we were mindful to be not redundant with the key points discussed.
Point 4: Is the RNA-seq data is submitted in any public domain. If submitted please provide accession IDs.
Response 4: The RNA-seq data were submitted to the NCBI BioSample database (https://www.ncbi.nlm.nih.gov/sra/SRP150148) as noted in results section (line 114) and the information was also provided in the Supporting Table S1.
Round 2
Reviewer 1 Report
Manuscript (ijms-400077) “Comparative transcriptome analysis reveals the role of fatty acid desaturases in preferential synthesis of α-linolenic acid in tree peony seeds” by Zhang et al. presents an interesting study about the high-throughput sequencing of the Tree peony transcriptome in seed.
However despite the large amount of work done and data presented, the methodology and results presents some inconsistencies and deficiencies. Mainly, authors must add more genes in the RTqPCR validation. Only three genes is not enough.
For these reasons, this manuscript is acceptable for publication in International Journal of Molecular Sciences AFTER A MODERATE/MAJOR REVISION.
The major points for the revision of the manuscript are:
In the Abstract and Introduction, the scientific name of tree peony must be incorporated (genus and species).
Objective of the work must be clearly specified in a separated paragraph excluding bibliography.
Plant material assayed must be better explained. In my opinion a clear phenotypic characterization is necessary of the two assayed species. By one side, in the Methodology sections section authors must include the phenotypic evaluation protocol. By other side, a new Table of Figure is required with the completed phenotypic evaluation. Authors must separate this information from the Figure 1. In addition, in the Results and Discussion section authors must include the phenotypic evaluation results as a separated section.
A new Figure with the fatty acid synthesis in the plastid should be incorporated.
RNA-Seq data and qPCR data from Figure 6 should be statistically compared. Author must incorporate a correlation coefficient of data between RNA-Seq and qPCR. In addition, I higher number of genes must be incorporated. Only three genes is not enough.
Figure 7 should be spliced in other two figures to increase its quality.
Reviewer 3 Report
Now, Manuscript can be accepted for publication.